# Population-Based Survival Analysis of Stage IVB Small-Cell Neuroendocrine Carcinoma in Comparison to Major Histological Subtypes of Cervical Cancer

Seiji Mabuchi [1,*,†], Naoko Komura [2,†], Tomoyuki Sasano [3], Mina Sakata [1], Shinya Matsuzaki [1], Tsuyoshi Hisa [1], Shoji Kamiura [1], Toshitaka Morishima [4] and Isao Miyashiro [4]

1   Department of Gynecology, Osaka International Cancer Institute, Osaka 541-8567, Japan; tsuyoshi.hisa@oici.jp (T.H.)
2   Department of Obstetrics and Gynecology, Kaizuka City Hospital, Osaka 597-0015, Japan
3   Department of Obstetrics and Gynecology, Osaka Saiseikai Nakatsu Hospital, Osaka 530-0012, Japan
4   Cancer Control Center, Osaka International Cancer Institute, Osaka 541-8567, Japan
*   Correspondence: seiji.mabuchi@oici.jp; Tel.: +81-6-6945-1181
†   These authors contributed equally to this work.

**Abstract:** The aim of the current study is to investigate the survival outcome of stage IVB SCNEC of the uterine cervix in comparison to major histological subtypes of cervical cancer. A population-based retrospective cohort study was conducted using the Osaka Cancer Registry data from 1994 to 2018. All FIGO 2009 stage IVB cervical cancer patients who displayed squamous cell carcinoma (SCC), adenocarcinoma (A), adenosquamous cell carcinoma (AS), or small-cell neuroendocrine carcinoma (SCNEC) were first identified. The patients were classified into groups according to the types of primary treatment. Then, their survival rates were examined using the Kaplan–Meier method. Overall, in a total of 1158 patients, clearly differential survival rates were observed according to the histological subtypes, and SCNEC was associated with shortest survival. When examined according to the types of primary treatments, SCNEC was associated with significantly decreased survival when compared to SCC or A/AS, except for those treated with surgery. In patients with FIGO 2009 stage IVB cervical cancer, SCNEC was associated with decreased survival when compared to SCC or A/AS. Although current treatments with either surgery, chemotherapy or radiotherapy have some therapeutic efficacies, to improve the prognosis, novel effective treatments specifically targeting cervical SCNEC need to be developed.

**Keywords:** cervical cancer; stage IVB; survival; SCNEC; population-based study

## 1. Introduction

Small-cell neuroendocrine carcinoma (SCNEC) of the uterine cervix is a rare pathological subtype of cervical cancer (0.9–1.5% of all invasive cervical cancers), with an annual incidence of 0.06 per 100,000 women [1–4].

SCNEC of the uterine cervix is highly aggressive with a high probability of hematogenous and lymphogenous metastases [5,6]. According to a previous report, the most common sites of first metastasis were the lung (42%) and lymph nodes (40%), followed by the liver (27%). [7]. Consistent with these findings, SCNEC of the uterine cervix is more likely to present at an advanced stage. According to a recent population-based study conducted in the USA, the stage distribution of SCNEC histology was 30.0%, 10.6%, 21.3%, and 29.9% for stage I, II, III, and IV, respectively, which were clearly different from those of SCC (44.3%, 18.8%, 20.2%, 9.8%) or A/AS (61.1%, 11.3%, 10.3%, 8.4%) histologies [2]. The incidence of stage IVB SCNEC according to the FIGO 2009 staging system has been reported to be roughly 25% [1,2].

Although the etiology of SCNEC of the uterine cervix, human papillomavirus (HPV) infections, is same as that of squamous cell carcinoma (SCC), adenocarcinoma (A), and

adenosquamous cell carcinoma (AS) [8], SCNEC has been associated with worse clinical outcomes compared to SCC or A/AS [2]. In patients with early-stage diseases (stage IB-IIA), the hazard ratio (HR) for death was reported to be 2.96 times higher for SCNEC compared to SCC [2]. In patients with locally advanced disease (stage IIB-IVA), the HR for death was 1.70 times higher for SCNEC compared to SCC [2]. With regard to stage IVB patients, with the marked advances in chemotherapy and immunotherapy, the survival of patients with an SCC or A/AS histology has been significantly improved in the last 25 years [9]. However, mainly due to its rarity, the prognosis of stage IVB SCNEC patients has not been fully investigated. Thus, its relative prognosis compared to that of SCC or A/AS remains unclear.

In order to address such a situation, in this population-based retrospective cohort study using the Osaka Cancer Registry data from 1994 to 2018, we retrospectively assessed the survival outcomes of FIGO 2009 stage IVB cervical SCNEC patients in comparison to cervical SCC or A/AS patients.

## 2. Materials and Methods

### 2.1. Data Source

This retrospective study was conducted using data obtained from the population-based Osaka Cancer Registry (OCR). The OCR is a full-longitudinal survey that collects information on the diagnosis, treatment, and survival of all cancers in Osaka Prefecture. The OCR has been in operation since 1962 [10].

The OCR records all new cancer cases in Osaka Prefecture that are identified by reports from medical facilities or the death certificate database. Patient data include age at cancer diagnosis, sex, date of cancer diagnosis, and date of death or the last follow-up. Tumor-specific data include site of cancer, disease extent, and histology. Disease extent was defined as follows: Localized (cancer confined to the original organ), Regional (cancer spread to immediately adjacent tissues and/or regional lymph nodes), and Distant (cancer metastasized to distant organs). The correspondence between the degree of disease extent and its FIGO 2009 classification is as follows: Localized—Stage I; Regional—Stage II-IVA; and Distant—Stage IVB. The histological subtype was classified in accordance with the morphology code of the *International Classification of Diseases for Oncology, 3rd Edition* (ICD-O3M). Squamous cell carcinoma (SCC); adenocarcinoma (A) including endometrioid carcinoma, serous carcinoma, clear cell carcinoma, mucinous carcinoma, or adenocarcinoma not otherwise specified (NOS); adenosquamous cell carcinoma (AS); and small-cell neuroendocrine carcinoma (SCNEC) were included in the analyses. Treatment data included treatment type (i.e., surgery, radiotherapy, and chemotherapy) and the hospital at which the cancers were diagnosed and treated. The OCR does not include information on patients' socioeconomic characteristics, performance status, pre-existing comorbidity, type of treatment and care after the initial treatment, follow-up details, hospital characteristics including surgeon volume and infrastructure, or causes of death. The vital status of the registered patients is routinely followed up at 3, 5, and 10 years from diagnosis using death certificates as well as official resident registries [10].

### 2.2. Study Population

The inclusion criteria were as follows: cases of uterine cervix neoplasia (C53—Malignant neoplasm of cervix uteri) registered in the OCR from 1994 to 2018, and patients who resided in Osaka at the time of diagnosis. No age restrictions were applied. Patients with carcinoma in situ, whose histology was a type other than SCC, A/AS, or SCNEC, those with multiple cancers, or whose extent of disease was unknown were excluded from the survival analyses. Accordingly, 1158 women with FIGO 2009 stage IVB cervical cancer displaying a SCC, A, AS, or SCNEC histology were analyzed for survival (Figure 1).

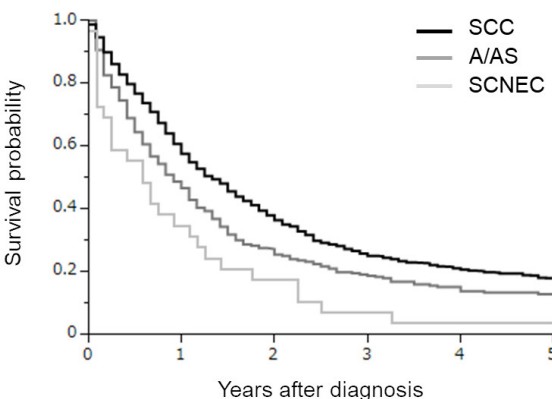

**Figure 1.** Kaplan–Meier estimates of overall survival of all cervical cancer patients included in the current study according to the histological subtypes. Clearly differential survival rates were observed according to the histological subtypes (SCC versus A/AS $p = 0.0001$; SCC versus SCNEC $p < 0.0001$; A/AS versus SCNEC $p = 0.0289$).

*2.3. Statistics*

Overall survival (OS) was defined as the time from the date of cancer diagnosis to the date of death or last follow-up visit. Comparisons of OS rates between groups were performed based on the Kaplan–Meier method, and the results were analyzed using the log-rank test. Comparisons of continuous data between groups were performed using Wilcoxon rank-sum test, Student's *t* test, or Kruskal–Wallis test, as appropriate. Proportions and frequency counts were compared between groups using Fisher's exact test or Chi-squared test, when applicable. All analyses were performed using JMP version 16.0 (SAS Institute, Cary, NC, USA) and a *p* value of <0.05 was defined as statistically significant.

**3. Results**

*3.1. Survival Outcomes of Stage IVB Cervical SCNEC, SCC and A/AS*

From 1 January 1994 to 31 December 2018, in total, 1158 stage IVB cervical cancer patients displaying a SCC, A, AS, or SCNEC histology were included in the analysis (Figure 1).

The clinicopathologic characteristics of the patients are presented according to year of diagnosis in Table 1. Of these, 842, 287, and 29 had SCC, A/AS, and SCNEC histologies, respectively. When compared, SCNEC was associated with significantly younger patients than SCC or A/AS. The predominant initial treatment included surgery for SCC, radiotherapy in A/AS patients, and radiotherapy or chemotherapy in SCNEC patients. In the survival analyses (Figure 1), clearly differential survival rates were observed according to the histological subtypes. The estimated 3-year survival rates in SCC, A/AS, and SCNEC patients were 25.5%, 18.8%, and 6.9%, respectively.

*3.2. Investigations According to the Type of Primary Treatment*

As can be seen (Table 1), of a total of 1158 patients, 186, 546, 229, and 179 were primarily treated with surgery (surgery group), radiotherapy (radiotherapy group), chemotherapy (chemotherapy group), and palliative care alone (palliative group), respectively.

In the surgery group, 97, 86, and 3 had SCC, A/AS, and SCNEC histologies. In the survival analyses (Figure 2A), a difference in OS was only observed between patients with SCC and A/AS histologies (SCC versus A/AS $p = 0.0099$; SCC versus SCNEC $p = 0.7935$; A/AS versus SCNEC $p = 0.3947$). The estimated 3-year survival rates in SCC, A/AS, and SCNEC patients were 44.9%, 28.4%, and 33.3%, respectively.

**Table 1.** Clinicopathological characteristics of stage IVB cervical cancer patients according to the histological subtypes.

| | | SCC (Total = 842) N (%) | A/AS (Total = 287) N (%) | SCNEC (Total = 29) N (%) | *p*-Values |
|---|---|---|---|---|---|
| Age | Median (Range) | 60 (23–100) | 59 (21–91) | 49 (31–87) | *p* = 0.0100 SCC vs. A/AS; *p* = 0.2260 A/AS vs. SCNEC; *p* = 0.0106 SCC vs. SCNEC; *p* = 0.0048 |
| | ≤39 | 76 (9.0) | 24 (8.4) | 7 (24.1) | |
| | 40–60 | 355 (42.2) | 131 (45.6) | 15 (51.7) | |
| | 61≤ | 411 (48.8) | 132 (45.0) | 7 (24.1) | |
| Primary Treatment | Surgery | 97 (11.5) | 86 (30.0) | 3 (10.3) | *p* < 0.0001 |
| | Radiotherapy | 460 (54.6) | 77 (26.8) | 9 (31.0) | |
| | Chemotherapy | 148 (17.6) | 72 (25.1) | 9 (31.0) | |
| | Palliative care alone | 123 (14.6) | 48 (16.7) | 8 (27.6) | |
| | Unknown | 14 (1.7) | 4 (1.4) | 0 | |

SCC, squamous cell carcinoma; A, adenocarcinoma; AS, adenosquamous cell carcinoma; SCNEC, small-cell neuroendocrine carcinoma.

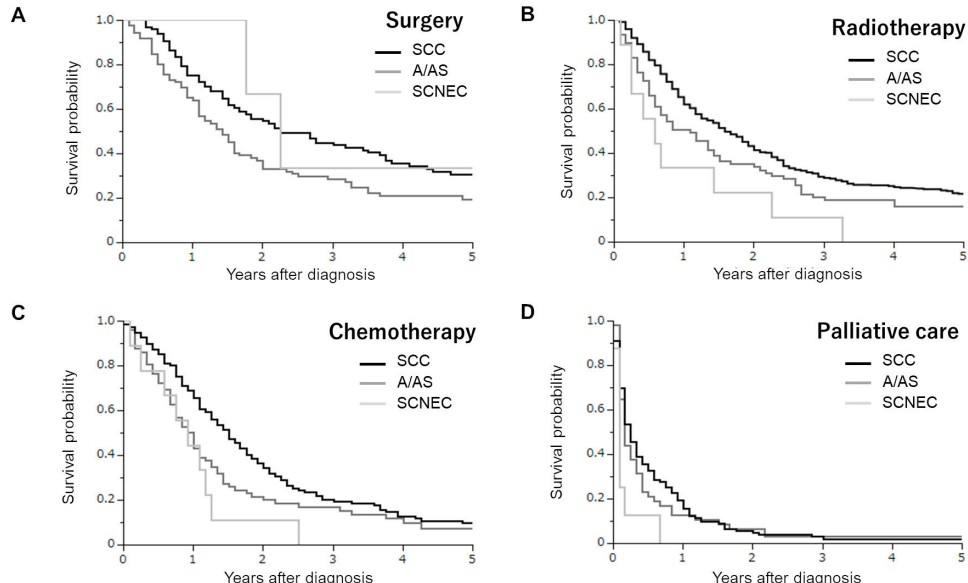

**Figure 2.** Kaplan–Meier estimates of overall survival of cervical cancer patients according to the type of initial treatment and histological subtypes. (**A**) Patients treated with surgery (SCC versus A/AS *p* = 0.0099; SCC versus SCNEC *p* = 0.7935; A/AS versus SCNEC *p* = 0.3947). (**B**) Patients treated with radiotherapy (SCC versus A/AS *p* = 0.0226; SCC versus SCNEC *p* = 0.0053; A/AS versus SCNEC *p* = 0.1513). (**C**) Patients treated with chemotherapy (SCC versus A/AS *p* = 0.0174; SCC versus SCNEC *p* = 0.0121; A/AS versus SCNEC *p* = 0.3109). (**D**) Patients treated with palliative care alone (SCC versus A/AS *p* = 0.5298; SCC versus SCNEC *p* = 0.0101; A/AS versus SCNEC *p* = 0.0456).

In the radiotherapy group, 460, 77, and 9 had SCC, A/AS, and SCNEC histologies. In the survival analyses (Figure 2B), patients with a SCNEC histology exhibited the lowest survival rate (SCC versus A/AS *p* = 0.0226; SCC versus SCNEC *p* = 0.0053; A/AS versus SCNEC *p* = 0.1513). The estimated 3-year survival rates in SCC, A/AS, and SCNEC patients were 29.3%, 20.3%, and 11.1%, respectively.

In the chemotherapy group, 148, 72, and 9 had SCC, A/AS, and SCNEC histologies. As shown in Figure 2C, patients with a SCNEC histology exhibited a similar survival rate to those with an A/AS histology, and had a significantly lower survival rate than those with a SCC histology (SCC versus A/AS *p* = 0.0174; SCC versus SCNEC *p* = 0.0121; A/AS

versus SCNEC $p$ = 0.3109). The estimated 3-year survival rates in SCC, A/AS, and SCNEC patients were 20.3%, 16.9%, and 0%, respectively.

In the palliative group, 123, 48, and 8 had SCC, A/AS, and SCNEC histologies. As shown in Figure 2C, patients with an SCNEC histology exhibited a significantly lower survival rate than those with an SCC or A/AS histology (SCC versus A/AS $p$ = 0.5298; SCC versus SCNEC $p$ = 0.0101; A/AS versus SCNEC $p$ = 0.0456). The estimated 1-year survival rates in SCC, A/AS, and SCNEC patients were 19.5%, 12.5%, and 0%, respectively.

### 3.3. Impact of Anti-Cancer Treatments in Stage IVB SCNEC Patients

Finally, we investigated the potential impact of anti-cancer treatments (either surgery, chemotherapy, or radiotherapy) in stage IVB SCNEC patients. For this purpose, the survival of SCNEC patients treated with palliative care alone (palliative group) was compared with those treated with some kind of treatment. As shown in Figure 3, all patients in the palliative group died of disease progression within one year (median OS of 1.0 months). Patients treated with some kind of treatment exhibited a significantly longer survival (median OS of 11.0 months) than those in the palliative group ($p$ < 0.0001).

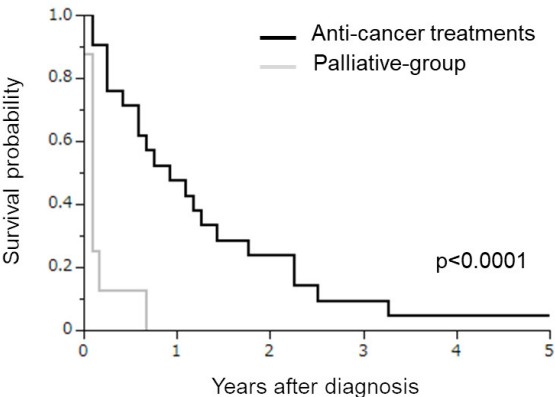

**Figure 3.** Kaplan–Meier estimates of overall survival of SCNEC patients treated with palliative care alone or anti-cancer treatments. Patients treated with palliative care alone exhibited a significantly shorter survival than those treated with anti-cancer treatments ($p$ < 0.0001).

### 4. Discussion

In this study, using data from the population-based cancer registry in Osaka Prefecture, we investigated the survival outcomes of FIGO 2009 stage IVB cervical SCNEC patients in comparison to cervical SCC or A/AS patients. We found that stage IVB SCNEC was associated with significantly decreased survival when compared to SCC or A/AS, except for the patients treated with surgical treatment. We also found that patients treated with anti-cancer treatments (either with surgery, chemotherapy, or radiotherapy) had a longer survival than those treated with palliative care alone (Figure 3 and Supplementary Figure S1). To the best of our knowledge, this population-based cohort study is among the largest to investigate the relative survival of stage IVB SCNEC in comparison to SCC and A/AS histologies in Japanese women with cervical cancer.

FIGO 2009 stage IVB distant metastases outside the pelvis included both visceral organ metastasis and lymph node metastases such as in the para-aortic or supraclavicular nodes. Due to the nature of the population-based Osaka Cancer Registry (OCR), we could not divide the patients into groups according to the site of metastasis. In patients with FIGO 2009 stage IVB cervical cancer with systemic metastasis, chemotherapy remains the mainstay of care. For patients with an SCC or A/AS histology, platinum-based combination chemotherapy combined with bevacizumab and/or Pembrolizumab has been the most effective chemotherapeutic regimen that is associated with a significant OS benefit [11,12]. However, there is no standard chemotherapy regimen for SCNEC based on good clinical evidence. Due to the histopathological similarity to small-cell lung cancer (SCLC),

the chemotherapy of SCNEC of the uterine cervix has been based on SCLC; etoposide plus cisplatin or irinotecan plus cisplatin has been most frequently employed regimen in Japan [13]. Globally, cisplatin plus carboplatin and cisplatin plus etoposide was the most commonly used regimen [14]. In patients with FIGO 2009 stage IVB cervical cancer with paraaortic lymph node metastasis who exhibit an SCC or A/AS histology, extended field radiotherapy with or without platinum-based concurrent chemotherapy has been employed as a potentially curative treatment. When such patients exhibit an SCNEC histology, multimodal treatments either with concurrent pelvic or extended-field chemoradiotherapy using etoposide plus platinum [15], the addition of pelvic radiotherapy after systemic chemotherapy [16], or surgical resection have been employed. However, as can be seen in Figure 1, the survival of SCNEC patients was significantly shorter than that of SCC or A/AS patients, indicating the limitation of current therapeutic strategies for cervical SCNEC, and the need for the development of novel effective treatments.

One possible strategy is the use of targeting agents. A next-generation sequencing study including 51 SCNEC patients revealed that genetic alterations were clustered in the RAS-MAPK pathway (42.86%), PI3K-AKT pathway (38.78%), p53 pathway (22.45%), and MYC family (20.41%) [17]. Similar results were obtained from another study [18]. These results indicate that SCNEC of the uterine cervix has a distinct genomic profile, and that novel agents targeting these pathways may have therapeutic effects against SCNEC of the uterine cervix.

Based on the promising clinical data obtained from cervical cancer patients with an SCC or A/AS histology [12], gynecologic oncologists might also expect immune checkpoint blockade therapies to be effective in cervical cancer patients with an SCNEC histology. Although one small study showed that three out of nine cases of cervical SCNEC were deficient in MMR [19], other larger studies indicated an MMR stable status in all cases tested [20,21]. On the basis of these results, from the point of MMR status, immune checkpoint inhibitions (ICIs) seem to be inactive in women with SCNEC. However, according to previous reports, the PD-L1 expression rate in cervical SCNEC was greater than 50% [21,22]. In fact, several case reports have shown that ICIs targeting the PD-1/PD-L1 axis are effective against cervical SCNEC [23,24]. In addition, other case series indicated a potential efficacy of dual CTLA-4 and PD-1 blockade against recurrent neuroendocrine cervical cancer [25,26]. The clinical activity of a bi-specific antibody targeting PD-1 and CTLA-4, cadonilimab (AK104), is currently being evaluated in a phase II clinical trial involving patients with recurrent high-grade neuroendocrine cervical cancer [27].

Considering the fact that surgically treated SCNEC patients exhibited higher survival rates than those in the chemotherapy group or radiotherapy group (Supplemental Figure S1), surgery can be considered a reasonable treatment in resectable FIGO 2009 stage IVB SCNEC patients. Another strategy that may improve patient's survival may be the addition of definitive radiotherapy after chemotherapy. A recent retrospective investigation demonstrated that the PFS and OS were longer when definitive radiotherapy was added after standard chemotherapy in patients with stage IVB cervical neuroendocrine carcinoma [28]. Moreover, in patients with paraaortic node metastasis, concurrent extended-field radiotherapy using etoposide plus platinum with or without neoadjuvant or adjuvant chemotherapy may also be a reasonable treatment. We hope the effects of the abovementioned strategies will be evaluated in future clinical trials that include a larger number of SCNEC patients.

The strength of our study lies in our analysis of long-term cancer registry data, the quality of which has been regarded as meeting the quality of the international standard [29]. However, we have to recognize the limitations of the current study. First, potential confounders, such as patients' socioeconomic characteristics, pre-existing comorbidity, performance status, postoperative adjuvant treatments, and hospital characteristics including surgeon volume and infrastructure, were not included in the OCR database. Similarly, the OCR database did not include details of the treatments (surgery, radiotherapy, or chemotherapy) or detailed pathological information such as tumor size, tumor grade, stromal invasion, nodal status, LVSI, tumor budding, cell nest size, SCNEC subclassifications,

immunohistochemical characteristics, or molecular profiles. Thus, we could not investigate the significance of a recently developed novel pathological grading system based on tumor budding and cell nest size [30,31] or molecular markers [31,32] in the current study. Second, we could not evaluate cancer-specific survival rates because the OCR database did not include the information on the cause of death. In addition, because the OCR used the SEER Summary Stage for registration and the tumor extent was only classified into Localized, Regional, or Distant categories, we could not evaluate the relative survival of stage IVB SCNEC compared to SCC or A/AS according to the size of the primary tumor, the site of metastatic tumors, or the number of tumor metastases. Third, although the Osaka Prefecture has a population of approximately 9 million people which is about one-thirteenth of the total population of Japan, the present study is not representative of the general population in Japan. Fourth, although SCNEC patients were younger and were more frequently treated with palliative care alone than SCC or A/AS patients (Table 1), due to the limited number of stage IVB SCNEC patients, the potential prognostic impacts of age or the treatment types could not be further investigated in the current study. Finally, although this study includes stage IVB patients assessed by the FIGO 2009 system, the included patients had been diagnosed between 1994 and 2018 and currently, patients with paraaortic node metastasis are classified as stage IIIC2 in the FIGO 2018 staging system and were not included in the study.

### 5. Conclusions

Using data from Osaka's large-scale, population-based cancer registry, we compared the survival rates of FIGO 2009 stage IVB SCNEC patients with those of SCC and A/AS patients who were diagnosed from 1994 to 2018. We found that the survival of FIGO 2009 stage IVB SCNEC patients was lower compared to that of SCC and A/AS patients. Although current treatments with either surgery, chemotherapy, or radiotherapy have some therapeutic effects, to improve the prognosis, novel treatments specifically targeting SCNEC of the uterine cervix need to be developed.

**Supplementary Materials:** The following supporting information can be downloaded at: https://www.mdpi.com/article/10.3390/curroncol30110682/s1, Supplemental Figure S1. Kaplan–Meier estimates of overall survival of SCNEC patients according to the type of initial treatment.

**Author Contributions:** Conceptualization, S.M. (Seiji Mabuchi); methodology, S.M. (Seiji Mabuchi), N.K. and T.S.; software, N.K.; formal analysis, N.K.; investigation, S.M. (Seiji Mabuchi), N.K. and T.S.; resources, S.M. (Seiji Mabuchi); data curation, T.M. and I.M.; writing—original draft preparation, S.M. (Seiji Mabuchi) and N.K.; writing—review and editing, S.M. (Shinya Matsuzaki), T.H., M.S. and S.K.; supervision, T.M. and I.M. All authors have read and agreed to the published version of the manuscript.

**Funding:** This research received no external funding.

**Institutional Review Board Statement:** The study was conducted in accordance with the Declaration of Helsinki, and approved by the Institutional Review Board of Osaka International Cancer Institute (approval number; 21144, date of approval; 11 November 2021).

**Informed Consent Statement:** Not applicable.

**Data Availability Statement:** The data presented in this study are available in this article (and Supplementary Material).

**Conflicts of Interest:** The authors declare no conflict of interest.

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
