# Peer review of "Population-Based Survival Analysis of Stage IVB Small-Cell Neuroendocrine Carcinoma in Comparison to Major Histological Subtypes of Cervical Cancer"

_curroncol, doi:10.3390/curroncol30110682_

Round 1

Reviewer 1 Report

Comments and Suggestions for Authors

Mabuchi et al. presented a population-based retrospective survival analysis of FIGO 2009 stage IVB small cell neuroendocrine cancer in comparison to other histological subtypes of cervical cancer using the Osaka Cancer Registry. They report that SCNEC were associated with decreased survival when compared to SCC or Adeno/Adenosquamous histologies except for those treated wit surgeries, and that stage IVB SCNEC patients treated with any modality had a longer survival than those treated with palliative care alone. 

My comments:

1. While there are several limitations to the study, including the lack of information about patient's pre-existing comorbidities, performance status, type of metastases, cause of death that could all influence the results of the study, I commend the authors for highlighting these limitations in their conclusions. I would however want to see some sort of control/sensitivity analysis where SCNEC would be matched to patients with SCC and A/AS by age (recognizing that SNEC patients tended to be younger) and other available demographics to see if the impact on survival or effect of therapies remains the same. 

2. Figure 2C: Should be "Chemotherapy" instead of "Radiotherapy" based on the figure legend 

 3.  line 154: correct "inly" with only

Comments on the Quality of English Language

The quality of the English is appropriate. Only a typo detected, to correct as above.

Author Response

Responses to Reviewer 1

We thank the reviewers for carefully reviewing our manuscript entitled “Population-based survival analysis of FIGO 2009 stage IVB small cell neuroendocrine carcinoma in comparison to major histological subtypes of cervical cancer”, and providing us with useful comments. Our manuscript was revised based on the reviewer’s comments.

Comment 1. While there are several limitations to the study, including the lack of information about patient's pre-existing comorbidities, performance status, type of metastases, cause of death that could all influence the results of the study, I commend the authors for highlighting these limitations in their conclusions. I would however want to see some sort of control/sensitivity analysis where SCNEC would be matched to patients with SCC and A/AS by age (recognizing that SNEC patients tended to be younger) and other available demographics to see if the impact on survival or effect of therapies remains the same. 

Response: Thank you for the reviewer’s thoughtful comment. As pointed, in our cohort, as shown in Table 2, SCNEC patients were younger than SCC or A/AS patients. In addition, the greater proportion of patients was treated with palliative care alone in SCNEC-group than in SCC or A/AS-groups (Table 2). These biased demographics could have influenced the results of the current study. We have indicated this as a limitation of the current study (lines 275-279 of the revised manuscript).

Comment 2. Figure 2C: Should be "Chemotherapy" instead of "Radiotherapy" based on the figure legend.

Response: I have corrected an error.

Comment 3. line 154: correct "inly" with only

Response: I have corrected an error.

Reviewer 2 Report

Comments and Suggestions for Authors

In the present study authors investigated the survival outcome of stage IVB SCNEC in comparison to major histological subtypes of cervical cancer. Manuscript is well written and data are clearly presented however, before accepatance i have a few suggestions to improove the overall quality of the paper:

1) Authors should refer to 2018 Figo staging system instead of FIGO 2009

2) A more detailed pathological assesment of all studied cases should be reported: tumor grade; LVSI, SILVA pattern of invasion...

3) Authors should state if SCNEC of their series were pure or intermixed with other histotypes (SCC, ADC)

4) Pathological and clinical prognostic factors for cervical cancer should be discussed (please include ref. Santoro A, Inzani F,et al. Recent Advances in Cervical Cancer Management: A Review on Novel Prognostic Factors in Primary and Recurrent Tumors. Cancers (Basel). 2023 Feb 10;15(4):1137.)

5) Differential diagnosis with other histotypes as well as novel markers fors cervical cancer should be also discussed (please include ref. Inzani F, et al. SATB2 is expressed in neuroendocrine carcinoma of the uterine cervix. Virchows Arch. 2022 Apr;480(4):873-877)

Author Response

Responses to Reviewer 2

We thank the reviewers for carefully reviewing our manuscript entitled “Population-based survival analysis of FIGO 2009 stage IVB small cell neuroendocrine carcinoma in comparison to major histological subtypes of cervical cancer”, and providing us with useful comments. Our manuscript was revised based on the reviewer’s comments.

Comment 1. Authors should refer to 2018 Figo staging system instead of FIGO 2009

Response: Thank you for the reviewer’s comment. As patients included in the current study were all diagnosed between 1994 to 2018, we could not use 2018 FIGO staging system. We have indicated this as a limitation of the current study (lines 279-282 of the revised manuscript).

Comment 2. A more detailed pathological assesement of all studied cases should be reported: tumor grade; LVSI, SILVA pattern of invasion.

Response: Osaka Cancer Registry data do not include detailed pathological information such as tumor grade or LVSI. So, we could not evaluate the potential effects of these factors in the current study. We have indicated this as an inherent limitation of the Osaka Cancer Registry data (lines 261-265 of the revised manuscript).

Comment 3. Authors should state if SCNEC of their series were pure or intermixed with other histotypes (SCC, ADC).

Response: In the current study, SCNEC cannot be further divided into pure SCNEC or SCNEC intermixed with other histotypes, as such detailed pathological information has not been included in the Osaka Cancer Registry data. We have indicated this as a limitation of the current study (line 264 of the revised manuscript).

Comment 4. Pathological and clinical prognostic factors for cervical cancer should be discussed (please include ref. Santoro A, Inzani F,et al. Recent Advances in Cervical Cancer Management: A Review on Novel Prognostic Factors in Primary and Recurrent Tumors. Cancers (Basel). 2023 Feb 10;15(4):1137.).

Response: In addition to conventional pathological risk factors such as tumor size, tumor grade, stromal invasion, nodal status or LVSI, the significance of a recently developed novel pathological grading system based on tumor budding and cell nest size could not be investigated in the current study. Moreover, in the current study, we could not be investigated the prognostic roles of recently proposed molecular markers. We have indicated this as a limitation of the current study (lines 261-267 of the revised manuscript).

Comment 5. Differential diagnosis with other histotypes as well as novel markers for cervical cancer should be also discussed (please include ref. Inzani F, et al. SATB2 is expressed in neuroendocrine carcinoma of the uterine cervix. Virchows Arch. 2022 Apr;480(4):873-877).

Response: Thank you for the reviewer’s thoughtful comment. We have included a short discussion on the recently proposed molecular markers by including the reference (lines 265-267 of the revised manuscript).

Round 2

Reviewer 1 Report

Comments and Suggestions for Authors

Thank you to the authors for addressing the reviewers comments and specifying these as further limitations to the study design and conclusions that can taken from it.

Author Response

Thank you for the reviewer’s thoughtful comment.